# Increasing efficiency of SVMp+ for handling missing values in healthcare prediction

**Yufeng Zhang**[1]*, **Zijun Gao**[1], **Emily Wittrup**[1], **Jonathan Gryak**[1,2,4], **Kayvan Najarian**[1,3,4,5]

**1** Department of Computational Medicine and Bioinformatics, University of Michigan, Ann Arbor, Michigan, United States of America, **2** Department of Computer Science, Queens College, City University of New York, New York, United States of America, **3** Department of Electrical Engineering and Computer Science, University of Michigan, Ann Arbor, Michigan, United States of America, **4** Michigan Institute for Data Science, University of Michigan, Ann Arbor, Michigan, United States of America, **5** Department of Emergency Medicine, University of Michigan, Ann Arbor, Michigan, United States of America

\* chloezh@umich.edu

**Data Availability Statement:** The MNIST+ data are collected from https://github.com/wenli-vision/svmplus_matlab, which is public and well-preprocessed. The heart failure dataset is a part of

## Abstract

Missing data presents a challenge for machine learning applications specifically when utilizing electronic health records to develop clinical decision support systems. The lack of these values is due in part to the complex nature of clinical data in which the content is personalized to each patient. Several methods have been developed to handle this issue, such as imputation or complete case analysis, but their limitations restrict the solidity of findings. However, recent studies have explored how using some features as fully available privileged information can increase model performance including in SVM. Building on this insight, we propose a computationally efficient kernel SVM-based framework ($l_2$-SVMp+) that leverages partially available privileged information to guide model construction. Our experiments validated the superiority of $l_2$-SVMp+ over common approaches for handling missingness and previous implementations of SVMp+ in both digit recognition, disease classification and patient readmission prediction tasks. The performance improves as the percentage of available privileged information increases. Our results showcase the capability of $l_2$-SVMp+ to handle incomplete but important features in real-world medical applications, surpassing traditional SVMs that lack privileged information. Additionally, $l_2$-SVMp+ achieves comparable or superior model performance compared to imputed privileged features.

## Author summary

Clinical problems often suffer from missing value issues, which require careful consideration. There are various approaches developed to tackle this problem, including imputation methods, but these methods have limitations. In this study, we introduced an efficient algorithm called $l_2$- SVMp+ to address missing values in important features using a partially available privileged information framework. Our approach offers a novel perspective for handling missing values by regarding them as privileged information to guide the training process. Our results indicate that (1) our proposed method outperforms the standard SVM, SVMp+; and (2) Our approach achieves comparable or superior

the PhysioNet Restricted Health Data, a freely-available medical research data platform. The dataset is available to qualified investigators which have been formally approved and under the terms of a data use agreement. https://physionet.org/content/heart-failure-zigong/1.3/ and contact contact@physionet.org for more information. The UCI heart disease dataset is publicly accessible on https://archive.ics.uci.edu/ml/datasets/heart+disease.

**Funding:** This material is based upon work supported by the National Science Foundation under Grant No. 1722801 and Grant No. 2014003. YZ received funding from National Science Foundation (NSF) Grant No. 2014003, ZG received funding from NSF Grant No. 1722801. EW and KN both received funding from NSF Grant No. 1722801 and Grant No. 2014003. The funders had no role in study design, data collection and analysis, decision to publish, or preparation of the manuscript.

**Competing interests:** The authors have declared that no competing interests exist.

performance to two commonly used imputation methods. This non-parametric approach offers a new direction for handling missing values and may potentially avoid imputation-related bias and overfitting in the model. With further testing and validation, our approach may lead to more accurate diagnoses and better treatment outcomes for patients.

## Introduction

Clinical Decision Support Systems (CDSS) rely heavily on machine learning algorithms to provide accurate predictions, but these algorithms are often challenged by missing values in Electronic Health Records (EHR) [1–3]. Given the complex nature of medical data, this is a common hurdle to algorithmic developments. For example, a patient in one condition might undergo multiple radiology scans while another patient receives specific lab tests, leading to missingness in both data modalities. Some other times, the information crucial for predicting patient outcomes could be missing due to various reasons such as changes in protocol and limited data access between institutions. Traditionally, researchers have resorted to methods like complete case analysis, dropping features, and imputation to handle missing data. Complete case analysis refers to discarding samples that have any missing values and restricting the study cohort to those with complete data [4]. Alternatively, all the features with missing values can be dropped from the analysis entirely. Despite of some successful applications, both methods may result in potential loss of valuable information or unnecessary reduction of sample size [5–7]. Instead of removing missing data, imputation replaces missing entries with predicted values based on the available data. Imputation has been increasingly and widely adopted in medical research [8]. When using this approach, however, it is important to note the potential reduction of model generalizability and reliability if obscuring the mechanisms of missing data [9].

Alternatively, a strategy called learning using partially available privileged information (LUPAPI) can make use of the important feature without imputation while addressing the issue of missingness [10]. This concept is derived from learning using privileged information (LUPI) [11]. Under the LUPI framework, high-quality features only available at the training stage but not during the testing stage are considered privileged information (PI) [12]. During the model training process, PI can play a teaching role to guide the model construction. Once trained, the model no longer has access to PI in the testing stage, just like the students taking exams without the teacher's assistance. This concept has been applied for object detection and image segmentation in computer vision tasks [13–15], but is also important for healthcare applications [10, 16, 17]. The LUPI framework allows models to use clinical information that may not be available at the time of prediction, such as lab tests or imaging results, but that can guide the model in learning the relationship between the main features and outcome. The LUPI paradigm was proposed for Support Vector Machines (SVMs) by Vapnik and Vashist [11, 12] which they called the SVM+ model. Apart from SVM, the LUPI paradigm can be applied to other machine learning algorithms and has already been applied to K-means [18] and Convolutional Neural Network for different tasks [13].

In practice, it is difficult to apply the LUPI scheme to clinical problems due to the sparse availability of PI across the training data. Assuming that only a portion of the patients has PI at the training stage, Sabeti proposed the LUPAPI paradigm [10] and provided an SVMp+ implementation. Unlike the standard LUPI framework, which required access to the PI for all training samples, LUPAPI guides the construction of the decision hyperplane of the SVMp+ using

slack functions defined on the partially available privileged feature space and slack variables for training samples without PI. Although the SVMp+ algorithm can handle sparse PI, it is computationally inefficient and incapable of modeling large datasets. This is due to constraining the slack variable and functions to be positive which doubles the number of dual variables and using a sequential minimal optimization (SMO)-style algorithm [19] which complicates working set selection.

In this paper, we present a novel extension of the LUPAPI paradigm, which resolve the limitations of the existing SVMp+ algorithm. Our proposed algorithm is based on $l_2$-loss [20], by which the number of dual variables is reduced. Besides, the derived dual representation is in the same form as one-class SVM, which could thus be solved using the existing state-of-the-art SVM solver LIBSVM [21] efficiently. We demonstrate the effectiveness of our algorithm through experiments on three datasets—digit recognition using the MNIST+ dataset, disease classification on the UCI heart disease dataset and readmission prediction on the PhysioNet Heart Failure dataset. Our results show that our proposed algorithm outperforms the SVMp + algorithm in terms of model performance and training time. Additionally, the readmission prediction task demonstrates that our algorithm achieves better results than imputation or feature elimination, making it a promising solution for handling missing values in EHR for CDSS applications.

## Methods

### Learning using partially available privileged information

In the LUPAPI paradigm, the additional privileged information is contained only in a subset of the training samples and is not available in the testing stage. We represent the training data as triplets $\{(\mathbf{x_i}, y_i, \tilde{\mathbf{x}}_\mathbf{i})\}$ for $i = 1, \ldots, m$ and pairs $\{(\mathbf{x_i}, y_i)\}$ for $i = m + 1, \ldots, n$, where $n$ and $m$ denote the sizes of the training dataset and the subset with privileged information respectively. For the $i$th training sample, $x_i \in \mathbb{R}^d$ represents the main features, $\tilde{\mathbf{x}}_\mathbf{i} \in \mathbb{R}^{\tilde{d}}$ represents the privileged features when they are available, and $y_i = \pm 1$ is the sample's label.

In the special case when $m = 0$, the original SVM model can learn from the training data $(x_i, y_i)$ a decision hyperplane of the form

$$\mathbf{w}^T\mathbf{x}_{\text{new}} - b = 0$$

where $w \in \mathbb{R}^d$ denotes the weight vector and $b \in \mathbb{R}$ denotes the bias, which could then be used to classify new data points $x_{\text{new}} \in \mathbb{R}^d$. By incorporating a feature map $\phi : \mathbb{R}^d \to \mathbb{R}^f$ and using the kernel matrix $K = \Phi^T\Phi$ where $\Phi = [\phi(x_1), \ldots, \phi(x_n)]$ denotes the data matrix of the feature vectors, the kernel SVM model can learn a decision boundary of the form

$$\mathbf{w}^T\phi(\mathbf{x}_{\text{new}}) - b = 0$$

where $w \in \mathbb{R}^f$, which is not necessarily a hyperplane. In the case when $m \geq 1$ and with the privileged information $\tilde{x}_i$, the model is guided to learn a new decision boundary with improved classification performance for new data points.

In [10], the SVMp+ was developed. The whole framework is based on SVM+ that [12] proposed, but there is one major difference. For those data points that have privileged information, the decision boundary is represented with the slack function $f(x) = \tilde{\mathbf{w}}^T\phi(\tilde{\mathbf{x}}) + \tilde{b}$, while for the data points which lack privileged information, the decision boundary is guided by slack

variable $\zeta$. The formula is shown below:

$$\min_{\mathbf{w},\mathbf{b},\tilde{\mathbf{w}},\tilde{b},\zeta} \frac{1}{2}\|\mathbf{w}\|^2 + \frac{\gamma}{2}\|\tilde{\mathbf{w}}\|^2 + C\sum_{i=m+1}^{n}\zeta_i + \tilde{C}\sum_{i=1}^{m}\left(\tilde{\mathbf{w}}^T\tilde{\phi}(\tilde{\mathbf{x}}_i) + \tilde{b}\right) \tag{1}$$

$$\text{s.t.}\begin{cases} y_i(\mathbf{w}^T\phi(\mathbf{x_i}) + b) \geq 1 - (\tilde{\mathbf{w}}^T\tilde{\phi}(\tilde{\mathbf{x}}_i) + \tilde{b}) & \text{for } 1 \leq i \leq m \\[2mm] \tilde{\mathbf{w}}^T\phi(\tilde{\mathbf{x_i}}) + \tilde{b} \geq 0 & \text{for } 1 \leq i \leq m \\[2mm] y_i(\mathbf{w}^T\phi(\mathbf{x_i}) + b) \geq 1 - \zeta_i & \text{for } m+1 \leq i \leq n \\[2mm] \zeta_i \geq 0 & \text{for } m+1 \leq i \leq n \end{cases}$$

where C, $\tilde{C}$ and $\gamma$ are hyperparameters greater than zero. The dual optimization problem can be formulated as:

$$\begin{aligned}
\mathcal{L} = \frac{1}{2}\|\mathbf{w}\|^2 \quad & + \frac{\gamma}{2}\|\tilde{\mathbf{w}}\|^2 + \frac{C}{2}\sum_{i=m+1}^{n}\zeta_i^2 + \frac{\tilde{C}}{2}\sum_{i=1}^{m}(\tilde{\mathbf{w}}^T\tilde{\phi}(\tilde{\mathbf{x}}_i)^{)2} \\
& - \sum_{i=1}^{m}\alpha_i(y_i(\mathbf{w}^T\phi(\mathbf{x}_i + b)) - 1 + \tilde{\mathbf{w}}^T\tilde{\phi}(\tilde{\mathbf{x}}_i) + \tilde{b}) \\
& - \sum_{i=1}^{m}\mu_i(\tilde{\mathbf{w}}^T\phi(\tilde{\mathbf{x_i}}) + \tilde{b}) \\
& - \sum_{i=m+1}^{n}\beta_i(y_i(\mathbf{w}^T\phi(\mathbf{x}_i) + b) - 1 + \zeta_i) \\
& - \sum_{i=m+1}^{n}v_i\zeta_i.
\end{aligned}$$

$$\text{s.t.}\begin{cases} 0 \leq \alpha_i \leq C & \text{for } m+1 \leq i \leq n \\[2mm] 0 \leq \alpha_i & \text{for } 1 \leq i \leq m \\[2mm] 0 \leq \beta_i & \text{for } 1 \leq i \leq m \end{cases}$$

The paper adopted the alternating SMO-style algorithm to optimize the dual cost function. Nevertheless, the working set is too complicated (at least 9 sets of maximally sparse feasible directions) and the dual variables interact. Therefore, the proposed algorithm is computationally slow.

## Solving the kernel $l2$-SVMp+ problem

In this part, we present a computationally efficient algorithm for solving kernel SVMp+. Based on the LUPAPI framework, we introduced $l_2$-loss into the kernel SVMp+ objective function and transforming it into one-class SVM. Here, we denote by $w$, $\tilde{w}$, $\phi$ and $\tilde{\phi}$ the weight vectors and feature maps for the main and privileged features taking values in the feature spaces $\mathbb{R}^f$ and $\mathbb{R}^{\tilde{f}}$ respectively. To simplify the formula, we absorb the bias terms into the weight vectors by introducing an auxiliary dimension to the weight vectors and feature maps.

Following the approach of Li in [20], we formulate the kernel $l_2$-SVMp+ model as the following optimization problem:

$$\min_{\substack{\mathbf{w}, \tilde{\mathbf{w}}, \rho \\ \zeta_{m+1}, \ldots, \zeta_n}} \frac{1}{2}\|\mathbf{w}\|^2 + \frac{\gamma}{2}\|\tilde{\mathbf{w}}\|^2 + \frac{C}{2}\sum_{i=m+1}^{n}\zeta_i^2 + \frac{\tilde{C}}{2}\sum_{i=1}^{m}(\tilde{\mathbf{w}}^T\tilde{\phi}(\tilde{\mathbf{x}}_i))^2 - \rho \tag{2}$$

$$\text{s.t.} \quad \begin{cases} y_i\mathbf{w}^T\phi(\mathbf{x_i}) \geq \rho - \tilde{\mathbf{w}}^T\tilde{\phi}(\tilde{\mathbf{x}}_i) & \text{for } 1 \leq i \leq m \\ y_i\mathbf{w}^T\phi(\mathbf{x_i}) \geq \rho - \zeta_i & \text{for } m+1 \leq i \leq n \end{cases}$$

where $\gamma, C, \tilde{C}$ are the hyperparameters of the model, $\zeta_i$ is the slack variable for the training samples without privileged information, and $\tilde{w}^T\tilde{\phi}(\tilde{x}_i)$ is the slack function for the training samples with privileged information defined in the privileged feature space. Therefore, the decision boundary learned from $(x_i, y_i)$ is further tuned by using the slack variables as well as slack functions during the training stage.

To solve the constrained convex optimization problem involved in the $l_2$-SVMp+ model, we first transform the primal problem to its Lagrange dual problem by using non-negative Lagrange multipliers. The Lagrangian function is given by:

$$\mathcal{L} = \frac{1}{2}\|\mathbf{w}\|^2 \quad + \frac{\gamma}{2}\|\tilde{\mathbf{w}}\|^2 + \frac{C}{2}\sum_{i=m+1}^{n}\zeta_i^2 + \frac{\tilde{C}}{2}\sum_{i=1}^{m}(\tilde{\mathbf{w}}^T\tilde{\phi}(\tilde{\mathbf{x_i}}))^2 - \rho$$

$$- \sum_{i=1}^{m}\alpha_i(y_i\mathbf{w}^T\phi(\mathbf{x_i}) - \rho + \tilde{\mathbf{w}}^T\tilde{\phi}(\tilde{\mathbf{x_i}}))$$

$$- \sum_{i=m+1}^{n}\beta_i(y_i\mathbf{w}^T\phi(\mathbf{x_i}) - \rho + \zeta_i).$$

By taking the partial derivatives with respect to the primal variables $w, \tilde{w}, \rho$ and $\zeta_{m+1}, \ldots, \zeta_n$ and setting them to zero, we obtain:

$$\frac{\partial\mathcal{L}}{\partial w} = 0 \rightarrow w = \sum_{i=1}^{m}\alpha_i y_i\phi(x_i) + \sum_{i=m+1}^{n}\beta_i y_i\phi(x_i)$$

$$\frac{\partial\mathcal{L}}{\partial\tilde{w}} = 0 \rightarrow \tilde{w} = \sum_{i=1}^{m}\alpha_i(\gamma I + \tilde{C}\tilde{\Phi}\tilde{\Phi}^T)^{-1}\tilde{\phi}(\tilde{x}_i)$$

$$\frac{\partial\mathcal{L}}{\partial\rho} = 0 \rightarrow 1 = \sum_{i=1}^{m}\alpha_i + \sum_{i=m+1}^{n}\beta_i$$

$$\frac{\partial\mathcal{L}}{\partial\zeta_i} = 0 \rightarrow \zeta_i = \frac{\beta_i}{C}$$

where $I$ denotes the identity matrix of the appropriate dimension, $\tilde{\Phi} = \left[\tilde{\phi}(\tilde{x}_1), \ldots, \tilde{\phi}(\tilde{x}_m)\right]$ denotes the data matrix of the privileged feature vectors, and the inverse matrix exists for all but a finite set of values of the ratio $\frac{\tilde{C}}{\gamma}$.

After combining $\alpha_i$ and $\beta_i$ into a single $n$-dimensional dual variable $\alpha = [\alpha_1, \ldots, \alpha_m, \beta_{m+1}, \ldots, \beta_n]^T$, we substitute the four equations back into the Lagrangian function to obtain:

$$\mathcal{L} = -\frac{1}{2}\alpha^T\left(\Delta_y K\Delta_y\right)\alpha - \frac{1}{2}\alpha^T\begin{bmatrix} \tilde{K}(\gamma I_m + \tilde{C}\tilde{K})^{-1} & 0 \\ 0 & C^{-1}I_{n-m} \end{bmatrix}\alpha$$

where $\Delta_y$ denotes the diagonal matrix with $y_1, \ldots, y_n$ along the diagonal, 0 denotes the zero matrix of the appropriate dimensions, and $K = \Phi^T\Phi$ and $\tilde{K} = \tilde{\Phi}^T\tilde{\Phi}$ are the kernel matrices for the main and privileged features respectively, and arrive at the Lagrange dual problem in matrix format

$$\min_{\alpha} \quad \frac{1}{2}\alpha^T\left(\Delta_y K\Delta_y + \begin{bmatrix} \tilde{K}(\gamma I_m + \tilde{C}\tilde{K})^{-1} & 0 \\ 0 & C^{-1}I_{n-m} \end{bmatrix}\right)\alpha$$

subject to the constraints that the entries of the vector $\alpha$ are all non-negative and sum to 1.

Therefore, the loss function is in the form of $\frac{1}{2}\alpha^T G\alpha$, which shares the same form as a kernel one-class SVM with kernel matrix $G$. This problem is considered a quadratic programming problem and can be solved by a standard SMO algorithm implemented in common SVM solvers such as LIBSVM. After calling the one-class SVM solver from LIBSVM to find the dual solution $\alpha_{\min}$, we could then compute the decision boundary as

$$\alpha_{\min}^T\Delta_y\Phi^T\phi(x_{\text{new}}) = 0.$$

Although the formula for the one-class SVM kernel matrix $G$ involves taking the inverse of a matrix, which is computationally expensive, the matrix in question is of moderate size $m \times m$. Moreover, the benefit of using highly efficient algorithms for both matrix inversion and solving one-class SVM makes the overall approach computationally efficient.

## Dataset

We utilized three datasets to illustrate the improvements of $l_2$-loss SVM+ under the LUPAPI paradigm on model performance for binary classification over alternative methods.

- MNIST+ dataset [12, 20]: A subset of the public MNIST+ dataset for classifying hand-written digits was used to benchmark algorithmic performance. Images for two digits: '5' and '8' were selected and split into training, validation, and test sets of 100, 4002, and 1866 images respectively. All images were resized to 10 by 10 pixels and then flattened into $100 - d$ vectors which were used as main features. The training images were appended by PI in the form of a textual description which was converted to a $21 - d$ vector.

- UCI heart disease dataset [22]: This dataset is composed of four databases: Cleveland, Hungary, Long Beach, and Switzerland, and initially included 76 attributes. In order to maintain consistency with other published papers, we used a subset of 14 attributes. One attribute identified whether the patient had heart disease, one attribute was for patient identification, and one described the data source. The remaining attributes were considered as main features, except for 'slope', which represented the slope of the peak exercise ST segment and was used as privileged information. After filtering out samples with missing main feature values, the dataset was reduced to 740 samples.

- PhysioNet Heart Failure dataset: This dataset was derived from the PhysioNet Heart Failure dataset. As part of the PhysioNet Restricted Health Data [23–25], PhysioNet Heart Failure dataset included EHR of 2008 patients admitted to Zigong Fourth People's Hospital with heart failure between December 2016 to June 2019. The patient cohort for our analysis was generated by selecting non-emergency patients whose discharging department and admission ward were both labeled as "cardiology". With the guidance of the cardiologist, 20 informative features are identified and used as the main features. After removing patients who had missing values for any of the listed features, the derived dataset contains 779 patients.

**Table 1. Demographic characteristics of patients in the PhysioNet heart failure dataset.**

| Features | Count | Percentage (%) |
|---|---|---|
| Gender | | |
| Male | 459 | 58.92 |
| Female | 320 | 41.08 |
| Age (years) | | |
| (21,29] | 3 | 0.39 |
| (29,39] | 5 | 0.64 |
| (39,49] | 30 | 3.85 |
| (49,59] | 39 | 5.01 |
| (59,69] | 147 | 18.87 |
| (69,79] | 324 | 41.59 |
| (79,89] | 202 | 25.93 |
| (89,110] | 29 | 3.72 |

The demographic statistics of the dataset are presented in Table 1 and clinical characteristics are listed in Table 2. Specifically, the dataset contains the left ventricular end-diastolic diameter (LVEDD) measurements for part of the patients, which is an important predictor of advanced heart failure [26–28].

**Table 2. Clinical characteristics of patients in the PhysioNet heart failure dataset.**

| Features | Units | Mean ± Std |
|---|---|---|
| Body temperate | Centigrade Scale | 36.41 ± 0.43 |
| Pulse | bpm | 82.65 ± 20.46 |
| Respiration | bpm | 18.74 ± 1.17 |
| SBP | mmHg | 128.56 ± 23.29 |
| DBP | mmHg | 75.27 ± 13.27 |
| MAP | mmHg | 93.03 ± 15.01 |
| WBC | $10^9/L$ | 6.84 ± 2.93 |
| HCT | % | 0.35 ± 0.069 |
| PLT | $10^9/L$ | 140.51 ± 55.74 |
| BMI | $kg/m^2$ | 21.89 ± 15.19 |
| BNP | pg/ml | 1202.03 ± 1310.52 |
| Creatine | umol/L | 108.79 ± 78.53 |
| Potassium | mmol/L | 4.01 ± 0.71 |
| Chloride | mmol/L | 101.84 ± 5.66 |
| Sodium | mmol/L | 138.46 ± 4.54 |
| Calcium | mmol/L | 2.31 ± 0.18 |
| Albumin | g/L | 37.07 ± 4.68 |
| NYHA class | class | III |
| LVEDD | mm | 53.79 ± 11.50 |

SBP = systolic blood pressure; DBP = diastolic blood pressure; MAP = mean arterial pressure; WBC = white blood cell; HCT = hematocrit; PLT = platlet; BMI = body mass index; BNP = brain natriuretic peptide; NYHA = New York Heart Association; LVEDD = left ventricular end-diastolic diameter

## Baselines

- Standard SVM: SMO-based *fitcsvm* [29] in MATLAB

- SVMp+: An implementation based on SMO algorithm. It solves the quadratic programming problem in Eq 1 [10].

- Mean imputation: The mean of non-missing values for every feature is computed to fill the missing values for all samples.

- Iterative Imputation: Iteratively estimating each feature based on the remaining features and then filling in the missing values. It is implemented in Python scikit-learn library [30].

## Evaluation metrics

The model performance was evaluated by F1 score, Area Under the Receiver Operating Characteristic Curve (AUROC) and Area Under Precision-Recall Curve (AUPRC). F1 is the harmonic mean of recall and precision, defined as

$$F1 \ \text{Score} = \frac{2 \times TP}{FN + 2 \times TP + FP}$$

where TP = true positive; FN = false negative; FP = false positive. F1 score reported here is the maximum based on a series of thresholds between 0 and 1.

## Experiments and results

### Digital recognition task on MNIST+

**Experimental design.** The original MNIST+ dataset was split into training, validation, and testing sets as shown in Fig 1A, and this same split scheme was used for all experiments. The best hyperparameter combinations for each experiment were selected based on performance on the validation dataset, and the model performance was evaluated using the testing dataset. In the standard SVM models, only the main features were used for training, while both the main and privileged features were utilized in the SVMp+ and $l_2$-SVMp+ models. To compare the performance of SVMp+ and $l_2$-SVMp+ when privileged information (PI) is only partially available in training, PI was randomly sampled under a specific seed to provide availability ranging from 50% to 90%. This sampling was repeated independently five times to yield five sets of PI under each availability level. Additionally, the sampling was performed to ensure robustness and provide statistical measures for the reported results.

**Model performance.** Fig 2 illustrates several evaluation metrics, including the area under the receiver operating characteristic (AUC), the area under the Precision-Recall curve (AUPRC), the F1 score and training time of standard SVM, SVMp+ and $l_2$-SVMp+. The training of standard SVM does not include any privileged information, therefore represented by PI ratio of 0%. The line reflects the change in mean values, while the bar represents the standard deviation across different models (when applicable). As a supplement, Table 3 lists clearer details of the mean values of the AUC, AUPRC, and F1 scores given different PI availability (Detailed model performance are shown in Supplementary S1 Table).

The standard SVM trained without any PI achieved an AUC of approximately 0.86. There is at least a 5% jump in AUC performance when PI is included in model training (6.7% for SVMp+ and 10.2% for $l_2$-SVMp+ with 50% PI ratio respectively). Specifically, the AUC performance of the $l_2$-SVMp+ is around 0.95 to 0.96. It slightly increases as the percentage of PI goes

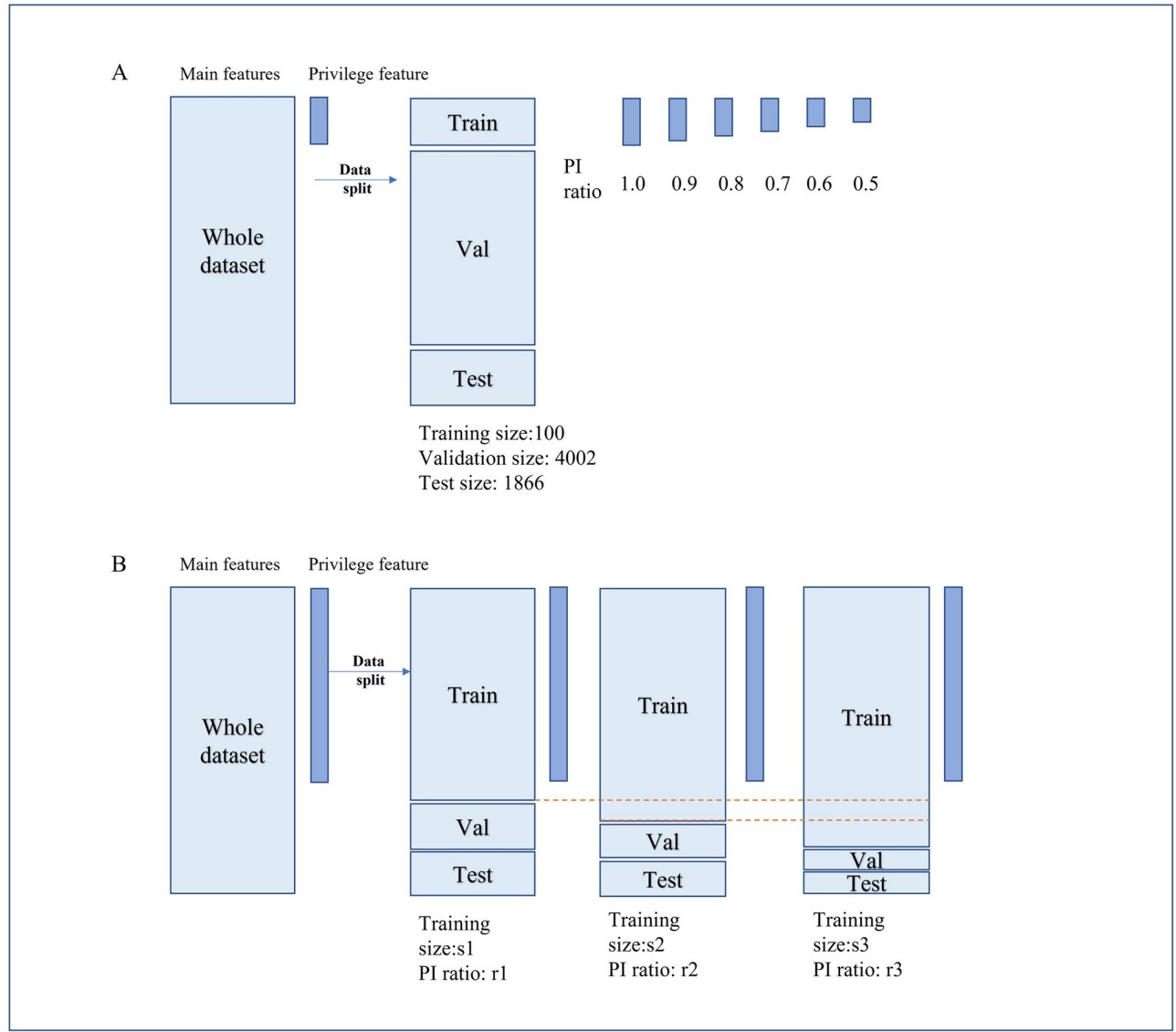

**Fig 1.** A: data split method for MNIST+; B: UCI heart disease and PhysioNet Heart Failure dataset split method: LVEDD or *slope* was regarded as privileged information (PI) in the analysis. All the patients with PI values were assigned to the training set and patients with missing PI values were randomly added to the training set with different probabilities to simulate different privilege information levels. The variables s1, s2, and s3 indicate distinct training sizes, whereas the variables r1, r2, and r3 correspond to their respective PI ratio levels.

up and reaches its highest value of 0.959 when the availability of PI is above 90%. As for the SVMp+ model, the best AUC of 0.92 is reached when 60% of the PI is present. Then, the performances have some fluctuations for different availability. But in general, the AUC values of the $l_2$-SVMp+ models are around 3% better than that in the SVMp+ model, given a fixed PI availability.

The AUPRC performances of the models follow a similar trend to that of the AUC. When using standard SVM, the AUPRC is roughly 0.85. Involving PI in training would contribute to 10% and 5% increases in performance with $l_2$-SVMp+ and SVMp+ models, respectively. When PI availability increases from 50% to 90%, the AUPCR of the $l_2$-SVMp+ model increases

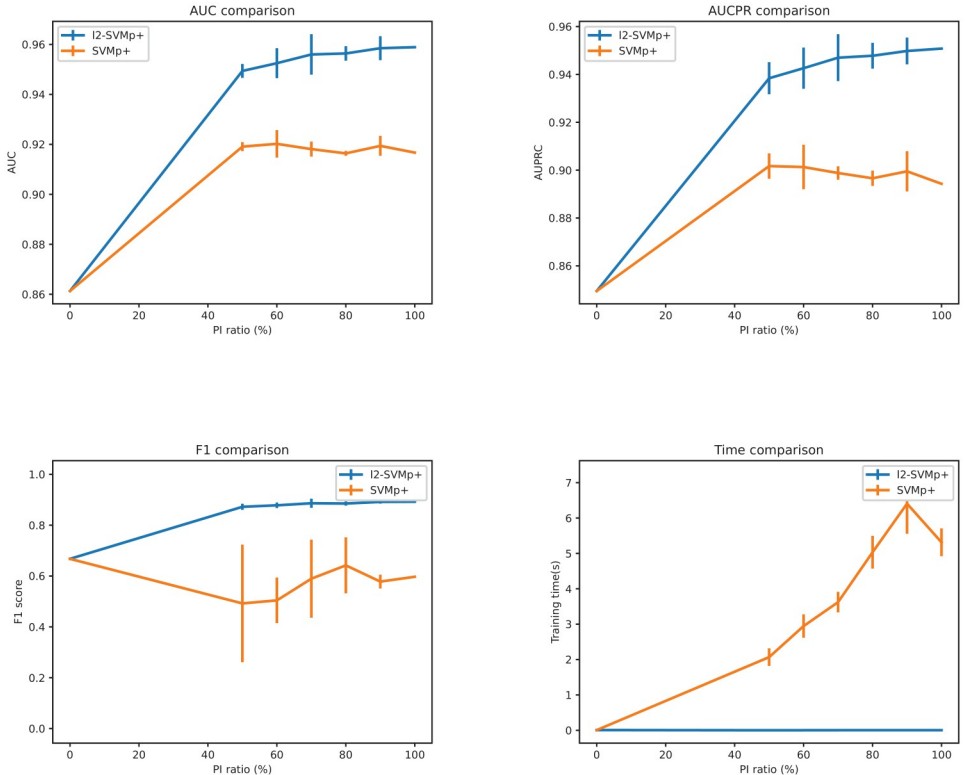

**Fig 2. Model performance on MNIST+: The x-axis is the ratio of privileged information.**

steadily from 0.983 to 0.95 and remains on the same level when PI is fully available. The SVMp+ models, on the other hand, show slight drops in AUPRC performance.

In terms of the F1 Score, the baseline SVM model gives a value of 0.668. When using $l_2$-SVMp+, the metric value increases to 0.872 under 50% availability and gradually grows to 0.892. However, the F1 Score of the SVMp+ model shows a decrease after adding PI, with the

**Table 3. Model performance comparison on MNIST+ dataset with ratio of privileged information varying.**

|  | PI Availability | AUC | AUPRC | F1 | Training time (s) |
|---|---|---|---|---|---|
| **Standard SVM** | 0% | 0.861 | 0.850 | 0.668 | 0.0068 |
| $l_2$-**SVMp+** | 50% | 0.949 | 0.938 | 0.872 | 0.0008 |
|  | 60% | 0.953 | 0.943 | 0.878 | 0.0020 |
|  | 70% | 0.956 | 0.947 | 0.886 | 0.0026 |
|  | 80% | 0.956 | 0.948 | 0.885 | 0.0029 |
|  | 90% | 0.959 | 0.950 | 0.892 | 0.0026 |
|  | 100% | 0.959 | 0.951 | 0.892 | 0.0033 |
| **SVMp+** | 50% | 0.919 | 0.902 | 0.492 | 2.0683 |
|  | 60% | 0.920 | 0.901 | 0.504 | 2.9464 |
|  | 70% | 0.918 | 0.899 | 0.590 | 3.6230 |
|  | 80% | 0.916 | 0.897 | 0.642 | 5.0336 |
|  | 90% | 0.919 | 0.900 | 0.578 | 6.4068 |
|  | 100% | 0.917 | 0.894 | 0.597 | 5.3134 |

mean value ranging from 0.49 to 0.64. In general, the $l_2$-SVMp+ model performs better on the F1 Score than the SVMp+ model, and the difference between the two could be more than 3%.

When PI availability ranges from 50% to 90%, Fig 2 illustrates the standard deviation across different experiments. With regards to the AUC and AUPRC, the SVMp+ models show either slightly smaller or equivalent standard deviations in comparison to the $l_2$-SVMp+ model. In contrast, the standard deviation of the SVMp+ model is much bigger than its $l_2$-SVMp+ counterpart in terms of the F1 Score, indicating that the $l_2$-SVMp+ model is more robust.

Moreover, the $l_2$-SVMp+ model shows clear superiority over the SVMp+ model regarding training times and is approximately one thousand times faster than the SVMp+ algorithm.

## Heart disease classification task on UCI dataset

**Experimental design.** Our objective is to develop a binary classifier utilizing the UCI dataset to identify patients with heart diseases. The missing rate of *slope* is up to 28.24% in the original dataset and to make the best use of it, all patients with *slope* were first assigned to the training set. Then, to simulate different levels of PI, the rest patients were randomly shuffled and then added into training or assigned to validation and test sets. The detailed split strategy is illustrated in Fig 1B. The shuffles result in three data groups; each has unique training, validation, and test set combinations. More specifically, within each group, the training set takes up 73%, 76%, and 80% of the original dataset, with PI availability of 98%, 94%, and 90% in training, respectively.

Standard SVM was carried out on the main features as a baseline without any privileged information, while SVMp+ and $l_2$-SVMp+ were performed on each data group with different PI availability. In addition, to compare imputation with our proposed method, we impute the *slope* variable to make it fully available for each data group with either (1) mean imputation or (2) iterative imputations and then treat the *slope* as part of the main feature to perform standard SVM.

In all experiments performed, the optimal hyper-parameter combination was selected based on the validation performance and the corresponding models were applied to test sets for final results.

**Model performance.** Table 4 displays the performance of five models on different data groups (Detailed model performances are shown in Supplementary S2 Table). Standard SVM trained with any imputation method and $l_2$-SVMp+ algorithm outperformed the standard SVM among all data groups regarding AUC, AUPRC and F1 score. In group 1, the standard SVM trained with iterative imputed PI achieved the highest AUPRC and F1 score. Nevertheless, the $l_2$-SVMp+ model obtained the best AUC (0.841), as well as a good AUPRC (0.608) and F1 score (0.685), which are comparable to the iterative imputation methods. In group 2, the $l_2$-SVMp+ model exhibited the best model performance with the highest AUC (0.844), AUPRC (0.640), and F1 score (0.692). For group 3, the iterative imputation methods demonstrated the best performance regarding AUC and AUPRC, although $l_2$-SVMp+ had similar results with a 1.1% decrease in AUC and a 1.7% decrease in AUPRC respectively. $l_2$-SVMp+ achieved the best F1 score with 0.694 in this experimental setup. SVMp+ only outperformed standard SVM in group 1 and group 2 with AUC increased by 1.22% and 2.86% respectively while performing worst in group 3. In addition, it took $\approx$ 1000 times slower than $l_2$-SVMp+ when using different training proportions.

The table also includes the AUC difference between the training and testing sets, measuring model fitting characteristics. The SVMp+ and $l_2$-SVMp+ models achieved the smallest AUC differences between the training and test sets in all three data groups, indicating good fitting characteristics.

**Table 4. Comparison of model performance on the UCI HF dataset.**

| Data Split | Metrics | | SVM w. Imputed *slope* | | *slope* as PI | |
|---|---|---|---|---|---|---|
| | | SVM | MeanImp[a] | MultiImp[b] | $l_2$-SVMp+ | SVMp+ |
| Group 1 73% for training | AUC | 0.818 | 0.837 | 0.840 | **0.841** | 0.828 |
| | AUPRC | 0.573 | 0.603 | **0.610** | 0.608 | 0.525 |
| | F1 score | 0.652 | 0.687 | **0.690** | 0.685 | 0.687 |
| | AUC diff* | 0.107 | 0.0984 | 0.0960 | 0.0570 | **0.0548** |
| | Training Time (s) | 0.0722 | 0.0182 | 0.0164 | 0.0424 | 58 |
| Group 2 76% for training | AUC | 0.804 | 0.837 | 0.844 | **0.844** | 0.827 |
| | AUPRC | 0.571 | 0.616 | 0.630 | **0.640** | 0.534 |
| | F1 score | 0.641 | 0.670 | 0.687 | **0.692** | **0.692** |
| | AUC diff* | 0.122 | 0.100 | 0.0935 | 0.0571 | **0.0568** |
| | Training Time (s) | 0.2221 | 0.0173 | 0.0164 | 0.0406 | 60 |
| Group 3 80% for training | AUC | 0.852 | 0.862 | **0.870** | 0.860 | 0.825 |
| | AUPRC | 0.606 | 0.623 | **0.635** | 0.624 | 0.512 |
| | F1 score | 0.673 | 0.674 | **0.694** | **0.694** | 0.664 |
| | AUC diff* | 0.058 | 0.074 | 0.0657 | **0.0395** | 0.0551 |
| | Training Time (s) | 0.078 | 0.0168 | 0.0173 | 0.0664 | 61 |

*AUC diff = Train AUC—Test AUC;

[a] Mean Imputation.

[b] Iterative Imputation

## Readmission prediction task on heart failure dataset

**Experimental design.** The task aims to predict patient readmission within six months based on the Heart Failure dataset introduced in ***Dataset*** Section. LVEDD, with a missing rate up to 36.3% in the original dataset, is considered PI for its important role in heart failure prediction. The data partition strategy and experimental setups are identical in the UCI heart disease classification problem as Fig 1B depicted. Shuffling leads to three data groups consisting of unique training, validation, and test sets. More specifically, within each group, the training set takes up 66%, 70%, and 75% of the original dataset, with PI availability of 97%, 92%, and 86% in training, respectively.

## Model performance

Table 5 presents the test performance of various models on different evaluation metrics (Detailed model performance on training and validation sets are shown in Supplementary S3 Table). As listed in Table 5, the standard SVM consistently achieves the lowest performance on AUC, regardless of the proportion of training data in the original dataset. However, The standard SVM trained with mean-imputed PI achieved slightly better AUC values than the standard SVM alone, while the standard SVM with iterative imputed PI achieved an even better AUC than the mean-imputed ones, with a 1%-2% increase. The $l_2$-SVMp+ model achieves the highest AUC performance among all models and training set proportions, with AUC values of 0.564, 0.565, and 0.596 when the training set proportion is 66%, 70%, and 75%, respectively. In terms of the AUPRC, the $l_2$-SVMp+ model is still the best-performing model, regardless of the data groups. In group 1, the standard SVM trained with mean-imputed PI obtained the worst performance. In groups 2 and 3, the standard SVM, the standard SVM trained with mean-imputed PI, and the one with iterative imputed PI showed increasing performance under a

**Table 5. Comparison of model performance on the PhysioNet HF dataset.**

| Data Split | Metrics | | SVM w. Imputed LEVDD | | LEVDD as PI | |
|---|---|---|---|---|---|---|
| | | SVM | MeanImp[a] | MultiImp[b] | $l_2$-SVMp+ | SVMp+ |
| Group 1<br>66% for training | AUC | 0.525 | 0.527 | 0.547 | **0.564** | N.A.** |
| | AUPRC | 0.547 | 0.546 | 0.564 | **0.573** | |
| | F1 score | 0.705 | 0.701 | 0.701 | **0.709** | |
| | AUC diff* | 0.303 | 0.449 | 0.429 | **0.158** | |
| | Training Time (s) | 0.076 | 0.025 | 0.025 | 0.022 | >48 hours |
| Group 2<br>70% for training | AUC | 0.534 | 0.550 | 0.562 | **0.565** | N.A.** |
| | AUPRC | 0.531 | 0.542 | 0.559 | **0.567** | |
| | F1 score | 0.696 | 0.690 | 0.692 | **0.701** | |
| | AUC diff* | 0.249 | 0.427 | 0.414 | **0.168** | |
| | Training Time (s) | 0.148 | 0.026 | 0.025 | 0.023 | >48 hours |
| Group 3<br>75% for training | AUC | 0.536 | 0.546 | 0.556 | **0.596** | N.A.** |
| | AUPRC | 0.544 | 0.551 | 0.561 | **0.598** | |
| | F1 score | 0.708 | 0.703 | 0.701 | **0.714** | |
| | AUC diff* | 0.239 | 0.430 | 0.420 | **0.153** | |
| | Training Time (s) | 0.063 | 0.030 | 0.030 | 0.039 | >48 hours |

*AUC diff = Train AUC—Test AUC;

** N.A.: no results obtained after 48 hours' training.

[a] Mean Imputation.

[b] Iterative Imputation

fixed training set proportion. Regarding the F1 score, $l_2$-SVMp+ remains optimal, while the standard SVM model achieves the second-best performance. The standard SVM models trained with mean-imputed or iterative-imputed PI data have the same level of F1 score and perform worse than the standard SVM model. It is worth noting that the SVMp+ model is unable to provide any performance metric due to its inability to converge in training after 48 hours, indicating poor scalability in implementation. The training times for the rest of the models are mostly less than 1 second.

Besides, The $l_2$-SVMp+ achieved the best generalizability regarding the AUC differences between the training and testing sets. The standard SVM gives the second smallest difference, while the two SVM models run on the imputed dataset have the same level of high AUC differences, indicating that they are likely to overfit the training set.

## Discussion

There are several strategies to address missingness in EHR data analysis, including discarding patients with missing values, dropping the features and data imputation; however, all will potentially compromise the study reliability when the missing mechanism is ignored. Initially designed for utilizing partially accessible privileged information, SVMp+ achieved success on the ARDS detection task. Nevertheless, its computational inefficiency poses a challenge to its application on a larger dataset.

In this work, an $l_2$-SVMp+ algorithm is proposed for efficiently solving kernel SVM in the case of learning using partially available privileged information. The improvements were made by applying an $l_2$ loss to the primal problem, simplifying its dual form and reducing the number of dual variables. This dual form is analogous to one-class SVM and can be efficiently solved by SMO solvers such as LIBSVM [21]. On MINST+, a dataset of a small scale in terms

of total training numbers, the $l_2$-SVMp+ displays comparable computational efficiency to the standard SVM model and is more than ten times faster than the previous SVMp+ implementation. This result is in line with previous findings from the $l_2$-loss kernel SVM algorithm [20]. Furthermore, when tested on the UCI heart disease dataset and the PhysioNet heart failure dataset, both of which are larger in scale compared to MINST+, the $l_2$-SVMp+ algorithm was able to converge within a second. In contrast, the SVMp+ algorithm exhibited much slower convergence and even failed to converge after 48 hours when tested on the PhysioNet dataset.

The $l_2$-SVMp+ algorithm also shows better performances compared to the SVMp+ algorithm on MINST+. Although both of the algorithms show superior testing outcomes on AUC and AUPRC compared to the standard SVM, the AUC, AURPC, and F1 Score achieved by $l_2$-SVMp+ are constantly better than those from the SVMp+ algorithm when the available percentage of privileged information range from 50% to 100%. This indicates that the $l_2$-SVMp+ algorithm itself is more effective than SVMp+ and can find an appropriate separation hyperplane more accurately with the help of privileged information. Noteworthy, the F1 score for the SVMp+ model on the MNIST+ dataset exhibits a significant decline in the mean value, coupled with a large standard deviation, when compared to the standard SVM baseline. Furthermore, while SVMp+ demonstrates improvement at training ratios of 73% and 76% on the UCI dataset, it shows a decrease in performance in comparison to the standard SVM model at a training ratio of 75%. These are indications of instability in the SVMp+ model. It may be caused by the alternating SMO-style algorithm and unoptimized implementation. But these problems were avoided in $l_2$-SVMp+ by directly using a well-established solver.

The $l_2$-SVMp+ algorithm outperformed the standard SVM algorithm in building a binary classifier for UCI heart disease patients and achieved comparable performance to the SVM model trained with imputed PI. Additionally, when predicting readmission using the PhysioNet Heart Failure dataset, the proposed algorithm consistently outperformed the standard SVM and SVM model trained with imputed LEVDD across all three data groups, demonstrating the best overall performance. Firstly, this result further supports our findings on MINST + that privileged information could improve the model's performance even when it is only partially available and $l_2$-SVMp+ remains effective in utilizing privileged information. Secondly, the results show empirical evidence that the LUPI paradigm could be superior to imputation methods when the missing information is important. Imputation relies on assumptions about the underlying data distribution and can result in biased estimates if those assumptions are incorrect or if the missing mechanism is unaware. This statement is supported by Table 5 and consistent with [31, 32], where imputation-based methods display larger AUC differences compared to the standard SVM model and LUPI-based $l_2$-SVMp+ model, indicating that they are prone to overfitting the training data.

Although the results from the dataset suggest that $l_2$-SVMp+ algorithm is either superior or on par with iterative imputation in handling missing data, there is currently a lack of theoretical support to make a concrete statement to that effect. Therefore, in future work, we will explore this hypothesis further by conducting theoretical analyses and experiments on large datasets to provide stronger evidence.

Future research should address the limitations present in the current work. The first limitation pertains to the lack of interpretability in clinical features. SHAP values, a method for explaining machine learning model outputs by calculating feature contributions to predictions, will be considered in the future to help interpret the model and rank feature importance, as suggested by Lundberg and Lee [33]. The second limitation concerns the fact that the test subset for the Heart Failure dataset is not a holdout but is randomly shuffled due to the high percentage of privileged information in the observations. Finally, the algorithm's application is

currently limited to only one partially available privileged information. To conduct a more comprehensive analysis, we will incorporate more variables with missing values.

## Conclusion

In this study, we introduced a highly efficient algorithm for solving the kernel SVMp+ problem. Our approach involves adding an $l_2$-regularizer to the original formulation, thereby converting the problem into a one-class SVM. This enables efficient and accurate optimization using an embedded SMO solver. We conducted extensive experimentation on three different tasks to evaluate the performance of our approach. Our results demonstrated that our method outperforms other common approaches for handling missing values, and showed superior efficiency and accuracy. In summary, the proposed algorithm presents a novel and highly effective solution for kernel SVMp+ in the context of missing values.

## Supporting information

**S1 Table. Performance of model on MNIST+ training, validation, and testing datasets with varying ratios of privileged information.**
(XLSX)

**S2 Table. Comparison of model performance on the UCI training, validation and testing dataset.**
(XLSX)

**S3 Table. Comparison of model performance on the HF training, validation and testing dataset.**
(XLSX)

## Author Contributions

**Conceptualization:** Yufeng Zhang.

**Data curation:** Yufeng Zhang.

**Formal analysis:** Yufeng Zhang.

**Funding acquisition:** Kayvan Najarian.

**Investigation:** Yufeng Zhang.

**Methodology:** Yufeng Zhang, Kayvan Najarian.

**Project administration:** Emily Wittrup, Jonathan Gryak, Kayvan Najarian.

**Resources:** Kayvan Najarian.

**Software:** Yufeng Zhang.

**Supervision:** Jonathan Gryak, Kayvan Najarian.

**Validation:** Yufeng Zhang, Zijun Gao.

**Visualization:** Yufeng Zhang.

**Writing – original draft:** Yufeng Zhang, Zijun Gao, Emily Wittrup.

**Writing – review & editing:** Yufeng Zhang, Zijun Gao, Emily Wittrup.

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
