## [Decision Letter · Decision Letter 0]

28 Apr 2023

PDIG-D-23-00046

Increasing efficiency of SVMp+ for handling missing values in healthcare prediction

PLOS Digital Health

Dear Dr. Zhang,

Thank you for submitting your manuscript to PLOS Digital Health. After careful consideration, we feel that it has merit but does not fully meet PLOS Digital Health's publication criteria as it currently stands. Therefore, we invite you to submit a revised version of the manuscript that addresses the points raised during the review process.

Please submit your revised manuscript within 60 days Jun 27 2023 11:59PM. If you will need more time than this to complete your revisions, please reply to this message or contact the journal office at digitalhealth@plos.org. Please include the following items when submitting your revised manuscript:

We look forward to receiving your revised manuscript.

Kind regards,

Mecit Can Emre Simsekler, Ph.D.

Academic Editor

PLOS Digital Health

Journal Requirements:

1. We ask that a manuscript source file is provided at Revision. Please upload your manuscript file as a .doc, .docx, .rtf or .tex.

3. Please provide separate figure files in .tif or .eps format only and remove any figures embedded in your manuscript file. Please also ensure that all files are under our size limit of 10MB.

Reviewers' comments:

Reviewer's Responses to Questions

**Comments to the Author**

1. Does this manuscript meet PLOS Digital Health’s publication criteria? Is the manuscript technically sound, and do the data support the conclusions? The manuscript must describe methodologically and ethically rigorous research with conclusions that are appropriately drawn based on the data presented.

Reviewer #1: Yes

Reviewer #2: Yes

2. Has the statistical analysis been performed appropriately and rigorously?

Reviewer #1: Yes

Reviewer #2: Yes

3. Have the authors made all data underlying the findings in their manuscript fully available (please refer to the Data Availability Statement at the start of the manuscript PDF file)?

Reviewer #1: Yes

Reviewer #2: Yes

4. Is the manuscript presented in an intelligible fashion and written in standard English?

Reviewer #1: Yes

Reviewer #2: Yes

5. Review Comments to the Author

Reviewer #1: Comments to Authors:

This study offers an efficient algorithm for handling missing values when utilizing Electronic Health Records (EHR) to develop Clinical Decision Support Systems (CDSS). The authors proposed algorithm for solving the kernel SVM-based framework (l2-SVMp+) by adding an l2-regularizer to the original formulation, converting the problem into a one-class SVM. Experiment on two different tasks validated the superiority of l2-SVMp+ over common approaches for handling missing values by showing high efficiency and accuracy.

Introduction:

Page 2 line 21: the authors should cite the paper about LUPAPI when they mentioned it for the first time.

Page 3 line 48: SMO algorithm was mentioned for the first time without any reference about it.

Methods:

Equation (1): there is no explanation about ξ, is it the same as ζ?

ζ as the slack variables or the correcting function allow misclassification with penalty hyperparameter C to determine decision boundaries. Then, could data with privileged information be given more weight which influence the decision boundary?

Dataset:

Page 6 line 91: After removing patients who had missing values, the derived dataset contains 848 patients. But, in Table 1, there are 779 patients based on gender (male 459 and female 320). Meanwhile based on age, 938 patients in total were analyzed. Please ensure the number of datasets used for this study.

Page 8: It is better to add figure of dataset split method for MNIST+ as well since it has different size of training, validation, and test sets.

Experiments and Results:

Figure 2 should be placed in page 9 before Table 3 to make it easy to read for the reader.

Page 9 line 105: The authors stated that there was a 5% jump in AUC performance when PI was included in model training. However, the AUC performance for SVMp+ was at least 6.7% while for l2-SVMp+ increased at least 10.22%.

Page 9 line 107: The authors mentioned that when using l2-SVMp+, the F1 value increases to 0.872 under 50% availability and gradually grows to 0.982. But, Table 3 showed that the highest F1 score was 0.892.

Page 11: Comparison of model performance in Table 4 showed that highest AUC value achieved by the proposed method was 0.596. AUC value tells how much the model is capable of distinguishing between classes. Based on a paper by Yang, et al [1], AUC value in 0.6-0.7 range means not good. Thus, the authors may elaborate more on F1 score and training time which was significantly reduced.

Discussion:

Page 12 line 123: The authors introduce the term SHAP for the first time and need to expand the abbreviation here.

Reference:

[1] Yang J, Liu X, Ai D, Fan J, Zheng Y, Li F, et al. (2015) PET Index of Bone Glucose Metabolism (PIBGM) Classification of PET/CT Data for Fever of Unknown Origin Diagnosis. PLoS ONE 10(6):e0130173. doi:10.1371/journal.pone.0130173.

Reviewer #2: The paper is interestingly discussing a way to deal with the datasets missing data by using the learn using partially available privileged information during models training which will not be available or to be used in the testing stage. The paper is well organized, written and presented.

I would suggest the authors to include the training, testing and validation results of the metrics used for the SVM (both imputation ways), SVMp+ and their proposed L2-SVMp+ models. As this would test the actual effect of the use of the LUPAPI.

I would suggest to add some illustrations for the easy understand the transition from training weights including PI to the testing weights without the PI.

6. PLOS authors have the option to publish the peer review history of their article (what does this mean?). If published, this will include your full peer review and any attached files.

**Do you want your identity to be public for this peer review?** For information about this choice, including consent withdrawal, please see our Privacy Policy.

Reviewer #1: Yes: Firda Rahmadani

Reviewer #2: No

---

## [Decision Letter · Decision Letter 1]

29 May 2023

Increasing efficiency of SVMp+ for handling missing values in healthcare prediction

PDIG-D-23-00046R1

Dear Miss Zhang,

We are pleased to inform you that your manuscript 'Increasing efficiency of SVMp+ for handling missing values in healthcare prediction' has been provisionally accepted for publication in PLOS Digital Health.

Best regards,

Mecit Can Emre Simsekler, Ph.D.

Academic Editor

PLOS Digital Health

Thanks for considering the comments.

Reviewer Comments (if any, and for reference):

Reviewer's Responses to Questions

**Comments to the Author**

1. If the authors have adequately addressed your comments raised in a previous round of review and you feel that this manuscript is now acceptable for publication, you may indicate that here to bypass the “Comments to the Author” section, enter your conflict of interest statement in the “Confidential to Editor” section, and submit your "Accept" recommendation.

Reviewer #1: All comments have been addressed

Reviewer #2: All comments have been addressed

2. Does this manuscript meet PLOS Digital Health’s publication criteria? Is the manuscript technically sound, and do the data support the conclusions? The manuscript must describe methodologically and ethically rigorous research with conclusions that are appropriately drawn based on the data presented.

Reviewer #1: Yes

Reviewer #2: Yes

3. Has the statistical analysis been performed appropriately and rigorously?

Reviewer #1: Yes

Reviewer #2: Yes

4. Have the authors made all data underlying the findings in their manuscript fully available (please refer to the Data Availability Statement at the start of the manuscript PDF file)?

Reviewer #1: Yes

Reviewer #2: Yes

5. Is the manuscript presented in an intelligible fashion and written in standard English?

Reviewer #1: Yes

Reviewer #2: Yes

6. Review Comments to the Author

Reviewer #1: The authors have addressed all comments raised in a previous round of review and even conducted additional experiment on other datasets. This manuscript is now acceptable for publication.

Reviewer #2: All comments were addressed, thanks.

7. PLOS authors have the option to publish the peer review history of their article (what does this mean?). If published, this will include your full peer review and any attached files.

**Do you want your identity to be public for this peer review?** For information about this choice, including consent withdrawal, please see our Privacy Policy.

Reviewer #1: No

Reviewer #2: No
